# Compression-sensitive smart windows: inclined pores for dynamic transparency changes

Haomin Chen[1], Gunho Chang[2], Tae Hee Lee[3], Seokhwan Min [2], Sanghyeon Nam[2], Donghwi Cho [4], Kwonhwan Ko[3], Gwangmin Bae[1], Yoonseong Lee [5], Jirou Feng [6], Heng Zhang[7], Jang-Kyo Kim[8,9], Jonghwa Shin [2], Jung-Wuk Hong[3] ✉ & Seokwoo Jeon [1] ✉

Smart windows, capable of tailoring light transmission, can significantly reduce energy consumption in building services. While mechano-responsive windows activated by strains are promising candidates, they face long-lasting challenges in which the space for the light scatterer's operation has to be enlarged along with the window size, undermining the practicality. Recent attempts to tackle this challenge inevitably generate side effects with compromised performance in light modulation. Here, we introduce a cuttlefish-inspired design to enable the closing and opening of pores within the 3D porous structure by through-thickness compression, offering opacity and transparency upon release and compression. By changing the activation mode from the conventional in-plane to through-thickness direction, the space requirement is intrinsically decoupled from the lateral size of the scatterer. Central to our design is the asymmetry of pore orientation in the 3D porous structure. These inclined pores against the normal direction increase the opaqueness upon release and improve light modulation sensitivity to compression, enabling transmittance regulation upon compression by an infinitesimal displacement of 50 $\mu$m. This work establishes a milestone for smart window technologies and will drive advancements in the development of opto-electric devices.

The era of smart windows is just around the corner. Implementing smart window technologies for temperature and lighting regulation in buildings can lead to a 50% reduction in energy consumption for building services[1–4]. As an important category in smart windows, light-

scattering smart optical membranes can be classified based on the stimuli that activate their optical modulation. For example, mechano-responsive membranes enable tunable transmission and redistribution of transmitted light through strain-controlled light scattering.

[1]Department of Materials Science and Engineering, Korea University, 02841 Seoul, Republic of Korea. [2]Department of Materials Science and Engineering, Korea Advanced Institute of Science and Technology, 34141 Daejeon, Republic of Korea. [3]Department of Civil and Environmental Engineering, Korea Advanced Institute of Science and Technology, 34141 Daejeon, Republic of Korea. [4]Thin Film Materials Research Center, Korea Research Institute of Chemical Technology, 34114 Daejeon, Republic of Korea. [5]Department of Nuclear and Quantum Engineering, Korea Advanced Institute of Science and Technology, 34141 Daejeon, Republic of Korea. [6]Department of Mechanical Engineering, Korea Advanced Institute of Science and Technology, 34141 Daejeon, Republic of Korea. [7]Department of Aeronautical and Aviation Engineering, The Hong Kong Polytechnic University, Hong Kong, China. [8]Department of Mechanical and Nuclear Engineering, Khalifa University of Science and Technology, Abu Dhabi, United Arab Emirates. [9]School of Mechanical & Manufacturing Engineering, University of New South Wales, NSW 2052 Sydney, Australia. ✉e-mail: j.hong@kaist.ac.kr; jeon39@korea.ac.kr

Compared with membranes responsive to electric/magnetic fields[5–12], heat[13–17], and light[18,19], mechano-responsive membranes hold significant promises owing to their energy-free operation plus many other advantages such as simple compositions, low cost, and rapid responses[20–24].

Various light-scattering membranes, known as scatterers, have been established with different functional configurations[6,25–32]. Two-dimensional (2D) scatterers achieve adaptive light scattering based on reversible flattening of surface wrinkles and cracks or infusion of surface roughness by liquid[26,30–45]. Although 2D scatterers can be easily fabricated, their optical contrast is lower than 60%, and the strain required for operation is higher than 30%, resulting in limited applicability. As an emerging alternative, three-dimensional (3D) nanocomposite scatterers composed of separable phases conformally contacting with each other have been developed[20,27,46–49] (Fig. 1a). Under tension, the delamination of these phases produces pores with a refractive index (~1.00) mismatched with that of the matrix (~1.40 in polydimethylsiloxane, PDMS), transforming the scatterers from transparent to opaque. These 3D scatterers, with numerous opened pores serving as scattering sites during stretching, exhibit strong multiple light scattering and transmittance contrasts typically higher than 70%[27]. Through a rational structural design optimizing strain distribution[20], the opening of pores is facilitated, considerably reducing the strain required for optical modulation.

Nevertheless, the foregoing remedy is far from a cure-all and has reached its bottleneck. The displacement required to activate the optical modulation is extensive and always coupled with the lateral size of the scatterer (Fig. 1a). For example, strains higher than 15% were inevitable to achieve appreciable contrasts[20,27,46]. Such a high in-plane strain means the scatterer must be stretched over 15 cm for transmittance modulation when used in a meter-sized window. Moreover, an additional space of more than 15% of the original window area was essential for the lateral elongation[24]. Recent improvements in contrast and the reductions in operational displacement and areal change sacrificed the benefits arising from simple composition and fabrication process when the fillers with Young's modulus

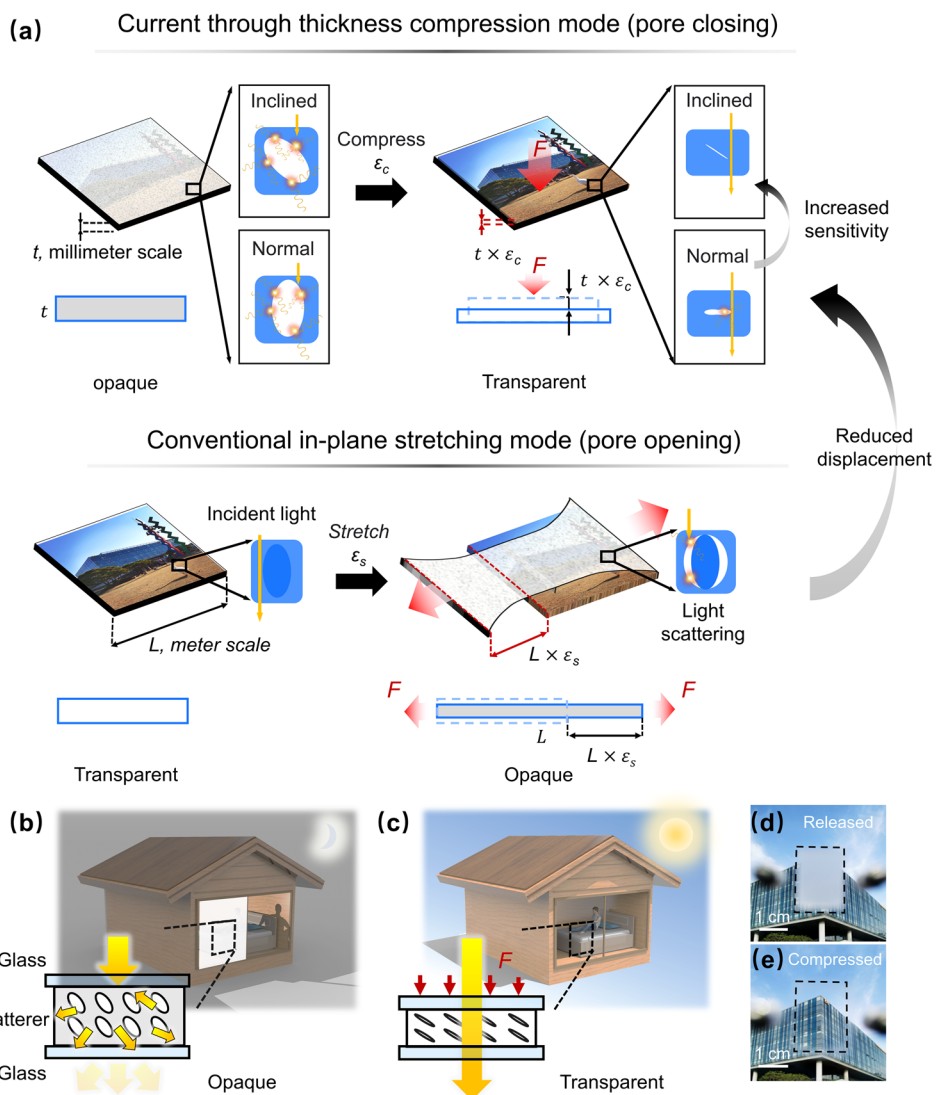

**(a)** Current through thickness compression mode (pore closing)

Conventional in-plane stretching mode (pore opening)

**Fig. 1 | Design concept of a 3D scatterer based on inclined porous structures loaded by through thickness compression.** Schematics illustrating different working mechanisms and operational displacements of the 3D scatterer loaded by (**a**) through-thickness compression and in-plane stretching. Schematics and digital images showing the compression-loaded 3D scatterer when used in smart windows under its (**b**, **d**) privacy-protection mode after release and (**c**, **e**) visible mode under through-thickness compression. *L* and *t* refer to the length and thickness of scatterers, $\varepsilon_c$ and $\varepsilon_s$ refer to the strain of compressing and stretching, *F* refers to the compressive or stretching forces. The shaded region and non-shaded region illustrate the opaque and transparent state of the scatterer, respectively. Thin black arrows indicate scales or magnified structures, thick black arrows indicate the logic flows, red arrows indicate the direction of force application, and golden arrows indicate the light propagation. Source data are provided as a Source Data file.

and refractive index distinct from those of matrices were introduced[20,27]. While the lateral strain was decreased to 5% by a shear-responsive scatterer based on the shielding effect of Fe$_3$O$_4$@SiO$_2$ nanochains[28], its maximum achievable transmittance and contrast were limited to 65% and 55%, respectively. Thus, a new design approach must be established to simultaneously ensure high visibility (transmittance > 90%), strong regulation of the transmittance (transmittance contrast > 90%), reasonable practicability (negligible operational displacement and areal change), and simple compositions with facile fabrication at a low cost.

In nature, cuttlefish are known for their ability to change color and pattern through the expansion and contraction of pigment-bearing chromatophores. Inspired by this ability, a potential solution to existing constraints is to manipulate the size of existing pores in a 3D structure by applying a through-thickness compression instead of creating new ones through in-plane strains. The reversible closing and opening of these pores enable the scatterer to be transparent when compressed and vice versa (Fig. 1a), where the pores play a similar role to the chromatophores. Despite its simplicity, this working mechanism intrinsically decouples the operational displacement from the lateral dimension of scatterers, restricting the operational displacement to the millimeter scale owing to the nature of compression applied through the thickness of the scatterer.

The performance of such a compression-driven scatterer relies on the optical density in the released state and sensitivity to compressive strains. Therefore, this study aims to develop 3D structures with submicron-inclined pores to develop an enhanced scatterer (Fig. 1a–e). The array of pores leads to strong multiple scattering of visible light, resulting in a high optical density in the released state and, thus, a high achievable transmittance contrast. With a tailored inclination angle, these submicron structures are fabricated using a 3D patterning technique based on a slanted exposure angle[50]. Under normal pressure, the local compressive strains are redistributed because of the presence of inclined pores, facilitating pore closure and increasing transmittance. The scatterers demonstrate high transmittance of 95% and unprecedented contrast of 94% with a marginal displacement during operation. Unlike the conventional scatterer based on an in-plane stretch mode, the one with a through-thickness compression mode allows localized modulation of transmittance by compressing only the area that needs to be transparent. The rational design of the compressive strain-responsive scatterer, featuring an array of submicron pores with an optimized inclined orientation, will not only shed insights into the development of next-generation smart windows[23], transparent displays[39], miniaturized optic devices[51], and strain visualizing sensors[52,53] but also propose a potential strategy for eliminating area expansion under compression.

## Results

### Compression-driven 3D scatterer based on inclined 3D structures

As an interference photolithographic technique known for producing large-area ordered 3D submicron structures by a single exposure step, proximity-field nanopatterning (PnP) can be used to fabricate the 3D porous structure for the proposed scatterer[54–56]. However, the advances in PnP have been focused on diversifying the symmetries, resolutions, and feature sizes using different designs of phase masks, while the control over the structural geometry remained largely unexplored. When the light is incident normal to the phase grating in the conventional PnP, diffracted beams are generated symmetrically to the normal direction, which interfere with each other and form periodic 3D patterns (Supplementary Fig. 1a). In contrast, when the incident light deviates from the normal direction, the horizontal component of the incident light may redirect the diffraction beams (Supplementary Fig. 1b), enabling the fabrication of an inclined 3D structure. The

diffraction angle of the $m$th order follows[55]:

$$\theta_m = \sin^{-1}\left(\frac{n_{mask}\sin\theta_i}{n_{pr}} + \frac{m\lambda}{n_{pr}P}\right) \tag{1}$$

where $\lambda$, P, $\theta_m$, $\theta_i$, n$_{mask}$, and n$_{pr}$ denote the incident wavelength, mask period, diffracted angle, incident angle, and the refractive indices of the phase mask and photoresist, respectively. Optical simulations were first conducted using the finite-difference time-domain method to verify the methodology. Two different PnP structures were studied using the same phase mask material with a 2D square array. The normal body-centered tetragonal or conventional PnP was converted to an inclined PnP by simply slanting the incident angle in the x-z plane against the normal direction (Supplementary Fig. 1c). When viewed from the side, the struts in the inclined or slanted PnP were tilted in the x-z plane but symmetric to the normal direction in the y-z plane. This is because the intensity distribution of the diffracted beams becomes asymmetric about the normal direction once the incident angle deviates from 0°, generating a slanted diffraction pattern.

To prepare the inclined 3D template, phase masks were carefully designed for slanted exposure (Supplementary Fig. 2). The prepared mask involved a 2D square array of gratings on the top surface and a substrate in the shape of a triangular prism (Fig. 2a). When the light was directed from the bottom, the monolithic design of the mask prevented any deviation of the incident angle before the light reached the gratings for phase modulation. The incident angle at the mask/photoresist interface was controlled by varying the angle of the prism-shaped mask. Figure 2b, c presents the simulated far-field diffraction patterns of normal and slanted exposure, consistent with experimental results (insets of Fig. 2b, c). The patterns became anisotropic when $\theta_i$ was changed from 0° to 45°, with the diffraction efficiency of the (1,0) order being considerably higher than that of the (−1,0) order. In addition, higher diffraction orders were produced along the x-axis. A series of interconnected porous structures with different slanted angles $\theta$ of 0.5°, 10.0°, 22.2°, 30.4°, and 40.3° were fabricated (Fig. 2e–j and Supplementary Fig. 3) by tuning the angle of the mask to 0°, 15°, 30°, 45°, and 65°. It is noted that their slanted angles and geometries matched well with the simulation results (Supplementary Fig. 4). The slanted angles of the struts measured based on fast-Fourier transform frequency domain patterns of the scanning electron microscope (SEM) images were generally smaller than the corresponding incident angles[57], attributable to refraction at the mask/photoresist interface. At $\lambda = 355$ nm, the refractive index of the NR photoresist was ~1.70, considerably higher than that of PDMS (~1.40). Plotted as a function of incident angle, the measured angles of the struts were consistent with the diffraction angles of the 0th order predicted by Snell's law (Fig. 2d), as shown in Eq. 1. These results indicate that in addition to the mask angle, the refractive indices of the mask and photoresist must be considered when fabricating inclined 3D structures.

It should be noted that the conformal contact between the photoresist and mask grating is crucial for exposure, particularly when the incident angle exceeds 45° (Supplementary Fig. 5). The experiment indicates that the required exposure dose increased with raising the incident angle: the optimized doses for the 0° and 65° incidence lights were 200 and ~1000 mJ/cm$^2$, respectively. This phenomenon is ascribed to the decayed diffraction intensity at higher angles, as confirmed by the optical simulation results (Supplementary Fig. 4).

The 3D porous PDMS light scatterers with different inclined angles were fabricated using the templates obtained above and can be activated by through-thickness compression (Fig. 2k). PDMS was selected as the matrix material considering its elasticity, non-adhesiveness, mechanical robustness, and high transparency to visible light. Because more than three steps are required to produce conventional in-plane

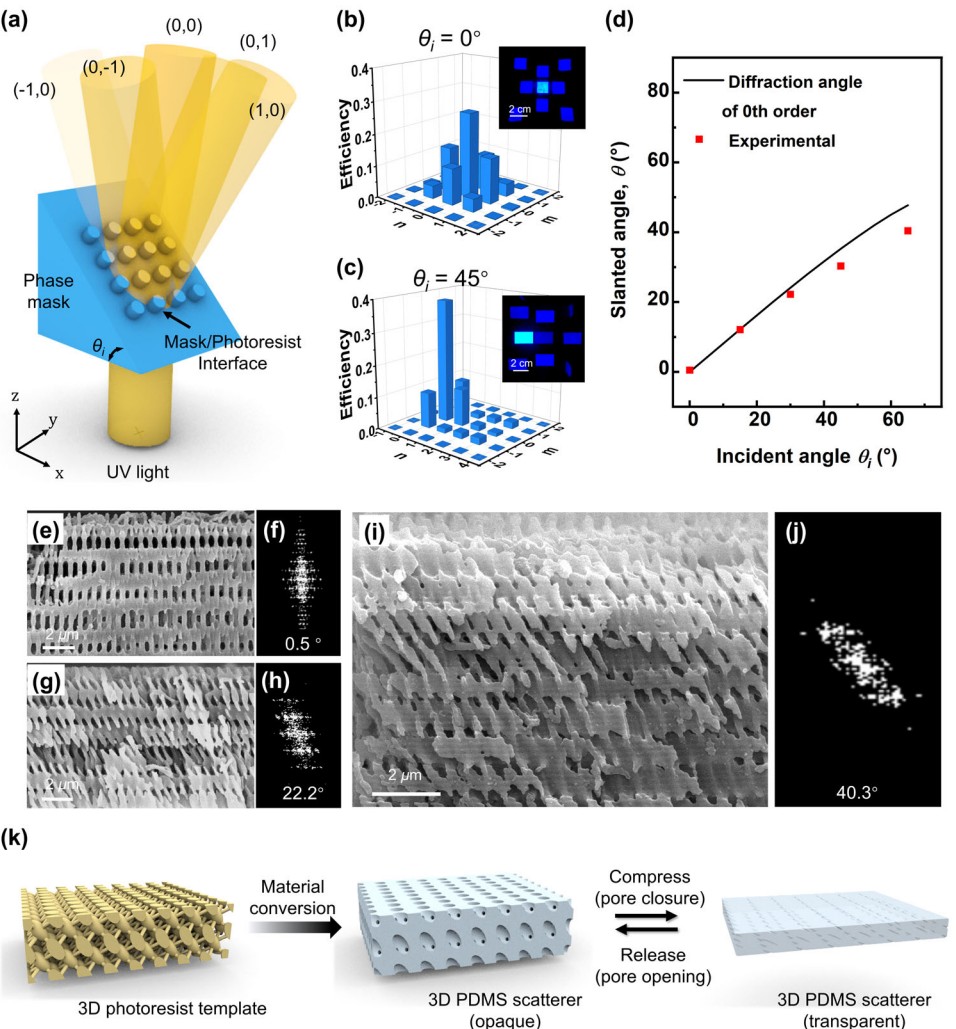

**Fig. 2 | Fabrication and characterization of 3D through thickness compression-mode scatterers. a** Schematic illustrating PnP using the prism-shaped phase mask. $\theta_i$ refers to the incident angle of exposure controlled by the mask. Golden pillars illustrate the propagation of light through the blue-colored phase mask. Visibility of the pillar illustrates the intensity of different far-field diffraction orders which are indicated by designators such as (0,0) and (1,0), etc. Simulated intensity distributions and digital images of the corresponding far-field diffraction patterns when **b** $\theta_i = 0°$ and **c** $\theta_i = 45°$. $n$ and $m$ indicate the order number of diffraction light along x and y axis. **d** Comparison between the measured angles of structural inclination and the predicted diffraction angles of the 0th order according to Snell's law. SEM images and the corresponding Fast-Fourier Transform frequency domain patterns of the inclined 3D structure prepared at incident angles of (**e** and **f**) 0°, (**g** and **h**) 30°, and (**i** and **j**) 65°. **k** Schematics illustrating one-step fabrication of 3D scatterer from the photoresist template. Source data are provided as a Source Data file.

stretching-mode scatterers with multiple separable phases (Supplementary Fig. 6)[20,27], the current through thickness compression-mode scatterers offer many benefits in terms of simple composition, facile fabrication, and low cost.

## Light-scattering performance

The light scattering performance of the 3D scatterers was evaluated by measuring changes in transmittance under compression at a wavelength of 550 nm (Fig. 3a). The scatterer was placed between two transparent substrates and compressed through thickness (Supplementary Fig. 7). The scatterer contained a functional layer of porous PDMS atop and a supporting layer of bulk PDMS at the bottom. The thickness of the functional layer and the complete scatterer were 12 μm and 0.6 mm, respectively (inset of Fig. 3a). Figure 3a shows the transmittance contrast, ΔT, of the scatterers with different slanted angles, as a function of applied compressive displacement, Δx. The scatterer without structural inclination presented a transmittance increase of ~76% under a relatively large Δx of 0.07 mm. In comparison, the

scatterer with a slanted angle of 40.3° exhibited a more pronounced increase in transmittance, with an extraordinary contrast of ~89% that saturated at a small Δx of 0.05 mm, equivalent to a compressive strain of 8.3% (Fig. 3a) and pressure of ~10 kPa (Supplementary Fig. 8). Supplementary Fig. 9 further illustrates the influence of a slanted angle on the sensitivity to compression, with the sensitivity calculated as the change in transmittance over displacement. The sensitivity consistently improved as the inclined angle increased from 0.5° to 40.3° for all compressive displacements. It is noticed that the sensitivity almost linearly increased in the initial displacement up to Δx = 0.05 mm when saturation occurred, followed by a gradual reduction in sensitivity with a further increase in displacement. Such a tendency may signify that the pores rapidly closed upon initial compression because of the low effective modulus of the porous structure. However, the sensitivity lessened after saturation owing to the density growth and, thus, the effective modulus.

The transmittance was continuously monitored for different compression displacements over a wavelength range between 400

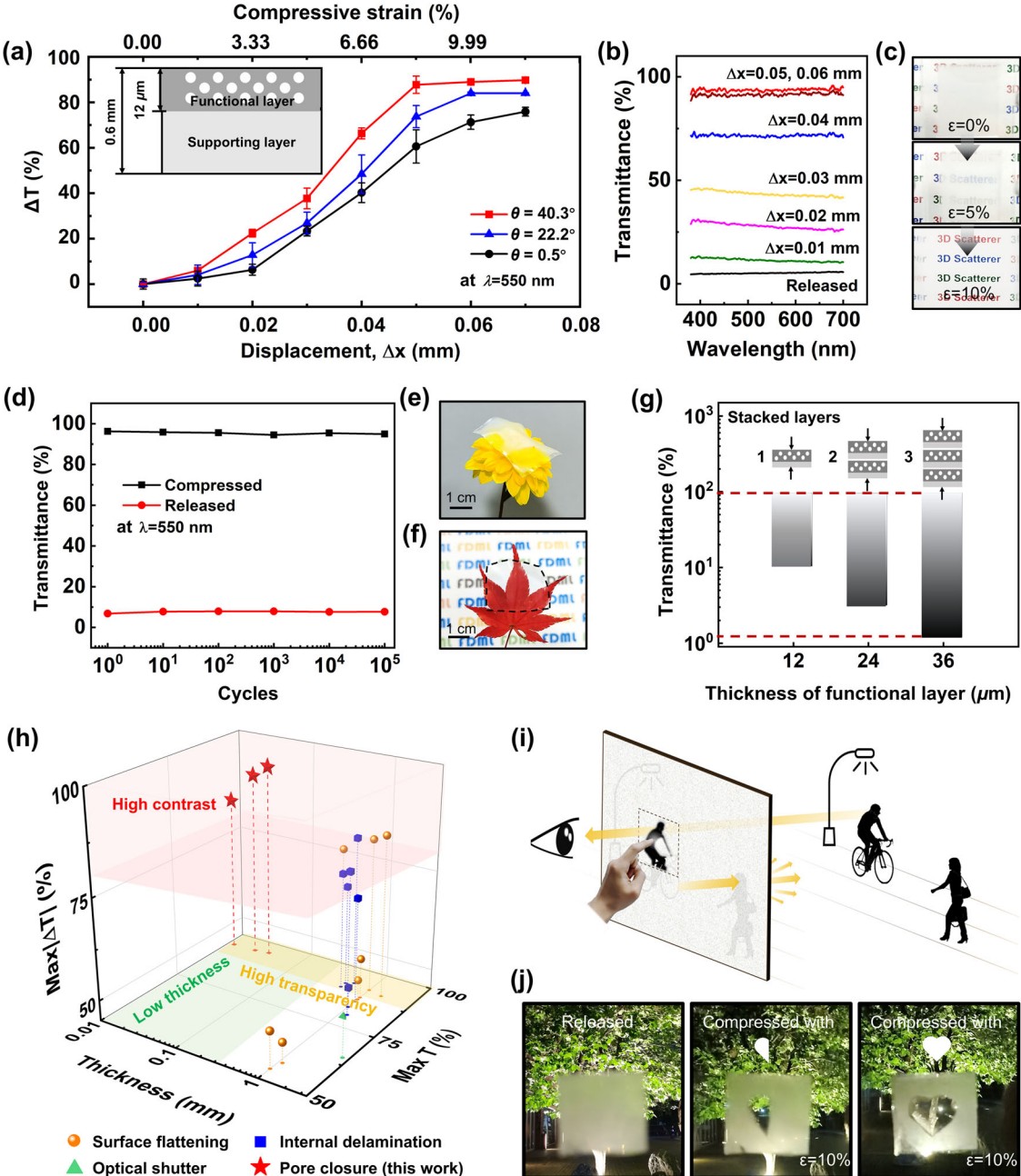

**Fig. 3 | Transmittance-modulating performance of the 3D scatterers.**
**a** Transmittance changes of the 3D scatterers with different inclined angles as a function of compressive displacement; Error bars represent the standard deviation. The inset shows a schematic illustrating the thicknesses of the porous functional layer and supporting layer in the 3D scatterer. Error bars indicate standard deviations; each data point is collected from 5 samples and the scattered point indicates mean of sample data. **b** Transmittance of the 3D scatterer with inclined pores at 40.3° measured at different compressive displacements. **c** Digital images showing the continuous transmittance modulation by the 3D scatterer. **d** Transmittance of

the 3D scatterer in the compressed and released states after cyclic tests. Digital images of the ultrathin 3D scatterer **e** placed on a flower and **f** on a leaf, making the behind letters invisible. **g** Transmittance modulation of the ultrathin scatterers containing different numbers of stacked layers. **h** Comparison of optical performance between our 3D scatterer and state-of-the-art mechano-chromic scatterers reported in the literature. **i** Schematic illustrating the 3D scatterer capable of selected-area optical modulation that can function as a peephole. **j** Digital images showing the optical modulation of different shapes by the 3D scatterer. Source data are provided as a Source Data file.

and 700 nm. For a typical 3D scatterer with an inclined angle of 40.3°, the transmittance measured without compression, i.e., in the released state, was as low as 6% (Fig. 3b), presenting completely blurred letters behind the scatterer (Fig. 3c). When the scatterer was compressed by 0.05 mm, the transmittance soared to 95% along with obvious letters. The transmittance values were almost independent of wavelength for all compressive displacements, reflecting the light scattering behavior governed by Mie scattering. In the case of total transmittance and total

reflectance, the modulation was approximately 20% but in opposite directions (Supplementary Fig. 10). The relatively high total transmittance of 77% in the released state can be rationalized by Mie scattering, which typically leads to a high proportion of forward-scattered light. During the optical modulation, area expansion below 5% was observed (Supplementary Movie 1), which can be rationalized by the inter-digitated pores in the 3D structure accommodating the deformations. A short response time of ~200 ms was noted (Supplementary Movie 1).

Moreover, the scatterer showed negligible changes in transmittance in both the released and compressed states even after 100,000 compression/release cycles at 0.06 mm (Fig. 3d) and 10 days of continuous compression (Supplementary Fig. 11). Such extraordinarily stable transmittance and structural durability have rarely been reported for existing mechano-responsive scatterers[21,24] and can be attributed to the simple and robust configuration, low operational compressive strains, and robust elasticity of PDMS.

Because optical modulation originates from the thin functional layer, the thickness of the scatterer can be reduced to tens of micrometers without degrading its performance. As a proof-of-concept, ultrathin 3D scatterers with a total thickness of only ~46 μm were prepared using spin-coating (Supplementary Figs. 12a, b). Notably, these scatterers were thinner than human hair[58]. The resulting ultrathin scatterer was light and flexible (Fig. 3e) but appeared optically dense in the released state, showing blurred letters at the back (Fig. 3f). The small thickness facilitated maximization of the contrast by conformally stacking multiple layers of scatterers. As shown in Fig. 3g and Supplementary Fig. 12c, the transmittance in the released state decreased from ~10% to ~1% as the number of stacked layers increased from 1 to 3. An approximate linear relationship was revealed between the optical density in the released state of the multi-layered scatterers and the thickness of the functional layer (Supplementary Fig. 12d), with a high coefficient of determination of 0.97. The linearity can be explained by the Lambert-Beer law[59,60], which relates the attenuation of light to the material thickness:

$$Optical\ density = -\log_{10} T = kt \qquad (2)$$

where T, k, and t are the transmittance, the coefficient indicating the attenuation capability per unit thickness, and the thickness of the material, respectively. Indeed, the linear relationship is invaluable when customizing the thickness of the functional layer depending on the desired opacity. A similar trend of increased opaqueness with the accumulation of stacked layers has been reported previously[36]. Nevertheless, a trade-off relationship existed between the maximum transparency and opacity in the existing frameworks because of light absorption in thickened materials and light scattering at the interfaces among the layers. The scattering by the interfaces can become more pronounced in the conventional scatterers based on in-plane tension, as the multilayer structure tends to delaminate when stretched because of tri-axial stresses[61]. Therefore, the transmittance contrast ΔT usually does not improve monotonically with the number of layers but tends to show a peak value at a certain number of layers. Interestingly, however, the compression-mode 3D scatterer developed here presented almost identical maximum transmittance in the compressed state among all samples with different numbers of layers ranging from 1 to 3 (Fig. 3g and Supplementary Fig. 12c). The negligible additional absorption can explain this observation due to the ultrasmall overall thickness and the elimination of interfaces by through-thickness compression. The improvements in both the opacity in the released state and the transparency in the compressed state gave an unprecedented contrast of ~94%, which is almost full-range modulation of transmittance. In principle, stacking multiple layers of scatterers is equivalent to preparing one scatterer with a functional layer of the same total thickness[20]. Therefore, increasing the thickness of the patternable submicron 3D structure would be a promising future direction to achieve improved scattering performance using one single layer of scatterer[50].

Given the same transmittance contrast, a thinner scatterer presents stronger modulation capacity and is preferred for microscopic optical devices. The scatterer's thickness is thus an important performance indicator in addition to contrast and transparency. The mechano-chromic performance is compared among the current 3D scatterers and the state-of-the-art scatterers reported in the literature.

Thanks to the compression activation mode based on the submicron-inclined porous structure, the current scatterers with different number of layers far outperformed other scatterers in terms of maximum transmittance contrast, maximum transmittance, and the total thickness (Fig. 3h)[20,27,32,34,36,38,40,43,46–48,62,63].

Another unique advantage of the scatterer based on through-thickness compression lies in its capability of selective local modulation. The transparency of some areas of the smart window incorporating the scatterer can be easily tailored by selectively compressing or releasing that region. Under the condition of unloading, the entire smart window remains opaque to protect indoor privacy. However, users can press any region of the scatterer to turn that specific area transparent, leaving the rest opaque (Fig. 3i). The transparent region can function as a temporary peephole, immediately returning opaque once the compression is removed. This freedom of local modulation is only possible for the scatterers using the through-thickness compression mode and can hardly be realized by the conventional mechano-responsive scatterers, where the tension or shear should be applied across the entire scatterer to change the transmittance. Although the region with tunable transparency could also be patternable by local treatment[20], the patterns had to be predetermined during fabrication and could not be modified afterward. The local modulation of transparency in the 3D scatterer was further demonstrated using different shapes of compressors (Fig. 3j). When the opaque scatterer was compressed locally in the shape of "half heart" or "full heart," only the selected area became transparent, revealing the trees in the background. Thanks to the remarkable modulation capability of normal transmittance along with the consistently high total transmittance, the scatterer can also function as diffusers for concentrated sun glare and improve the overall indoor illumination conditions during daytime[20]. Apart from their potential use in smart windows, the unique capability of the 3D scatterers to modulate local transparency may bring possibilities to developing transparent displays by rationally integrating the scattering membrane with arrayed actuators.

## Mechanisms of improved sensitivity

The deformation and transmittance of the 3D scatterer were simulated using finite element analysis to scrutinize the mechanism underlying the improved sensitivity in the inclined porous structure. The model was built to contain the porous component on top of the 40 times thicker solid component (Fig. 4a, whose details are given in the Supplementary Materials Supplementary Figs. 13a and 14). Models were built with three different orientations of the ellipsoidal pores, $\theta = 0°$, 20°, and 40°, against the thickness direction for the porous component. The three models presented almost identical strains in the porous component for given total strains regardless of pores' inclined angles (Fig. 4b), which could be explained by the comparable porosity and, thus, the analogous modulus of the porous component. However, it was revealed that the relative pore volume, $V_{norm}$, decreased more rapidly when $\theta$ rose from 0° to 40° (Fig. 4b). To clarify the reasons behind this behavior, the strain distributions in the functional layers with different $\theta$ were compared (Fig. 4c). In the model with vertically aligned pores, the strain distribution at 4% total strain was uniform, as indicated by large red and orange regions with unclear boundaries. When the compression was applied along the pore length, the pores tended to expand transversely, deforming into a more oval shape (Supplementary Fig. 15a). In contrast, the strains concentrated at the edges of the inclined pores with $\theta = 40°$, exhibiting a sharp contrast between the red and green colors. The large strain concentration is attributed to the reduced local compressive modulus at the thinner struts surrounding the pores compared to the thicker struts around the vertically aligned pores. Therefore, the inclined pores were more easily closed under compression of the same load than the vertically aligned pores, improving compression sensitivity and increasing the

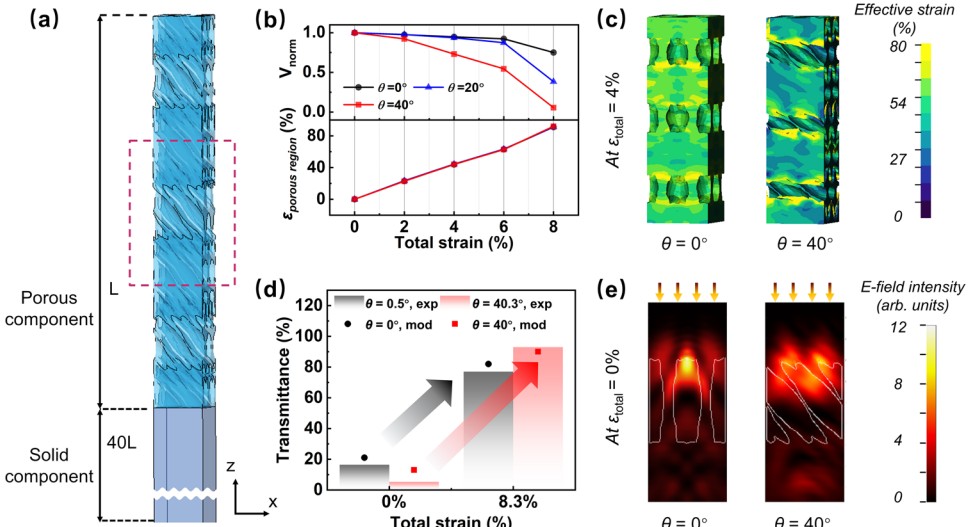

**Fig. 4 | Multiphysics simulation of the 3D structures with different pore inclinations. a** Models used in the multi-physics simulation. L and 40 L are the length of the porous and solid components. Porous component is disproportionately magnified to reveal the pore array. Red dashed box indicates the part of porous component chosen for optical simulations. **b** Strains in the regions with pores and normalized pore volume of the models with different pore inclined angles as a function of total strain. **c** Effective strain distributions of the models with inclined pore angles of 0° and 40° at 4% total strain. **d** Comparisons between the transmittance values of the models with inclined pore angles of 0° and 40° measured/predicted at 0% and 4% total strains by experiments and simulations. **e** Intensity distributions of scattered electromagnetic waves of the models in the released state. The white lines indicate the boundary between the pores and PDMS. Source data are provided as a Source Data file.

inclination angle. Moreover, the concentrated strains were asymmetric against the inclined pores and drove the pores to first rotate toward the horizontal direction, further facilitating the pore closure (Supplementary Fig. 15b and Supplementary Movie 2).

Optical simulations were conducted to analyze the light scattering behavior of the scatterers in both the released and compressed states. The middle layer of the porous component (red dotted box in Fig. 4a) was chosen for the study to avoid any boundary effect arising from the top and bottom. When an incident light was vertically shed from the top, the model with inclined pores at 40° presented a low transmittance of 13%, whereas the model with $\theta = 0°$ showed a transmittance of 21% (Fig. 4d). According to the intensity distribution of the scattered electromagnetic field, the field concentrated along the interface between the air and PDMS (Fig. 4e), verifying the critical role of pore/PDMS interface in scattering the incident light. In contrast to the symmetric intensity distribution in the model with vertically aligned pores, the asymmetry in the model with inclined pores led to off-normal transmission of light. The scattering cross-section represents the probability that the structure will deflect light in a specific direction, indicating the structure's overall effectiveness in scattering light. As shown in Supplementary Fig. 16, the model with inclined pores presented a more prominent scattering cross-section than the counterpart with vertically aligned pores throughout the visible spectra. Meanwhile, the multiple scattering boundaries, thanks to the overlapped edges of the slanted pores in the z-direction of the model, may also contribute to light scattering, accounting for the lower transmittance in the released state. When a sizeable compressive strain of 8.3% was applied, equivalent to a displacement of 0.05 mm for 0.6-mm-thick real samples, the transmittance of the model with inclined pores soared to 90% while the other model with vertical pores increased to only 82%, consistent with the experimental results (Fig. 4d). In summary, the multiphysics simulation confirmed that the significant optical contrast of the inclined porous structure may be attributed to the low initial transmittance in the released state and the high sensitivity of pore closure to compression. Future directions could include designing optimized geometries, such as auxetic structures, for efficient pore closure and extending 3D patterning techniques to realize these structures through inverse calculations.

## A potential strategy to eliminate large area expansion

In the application scenarios of mechano-chromic membranes, such as smart windows, the area expansion of membranes during operation is crucial, as it indicates how much space is needed for the installation and operation[24]. The frame dimensions must accommodate the membrane dimensions in both the released and the stretched states to ensure the membrane operates effectively within a fixed frame. Traditional scatterers activated by in-plane tension need large extra space for operation, exhibiting area expansion exceeding 10%. The additional space requirement (ASR) of traditional scatterers under uniaxial tension can be expressed as:

$$ASR = \frac{ab(1 + \varepsilon_{tension}) - ab}{ab} = \varepsilon_{tension} \quad (3)$$

where $a$ and $b$ denote the original lateral dimensions of the scatterer, and $\varepsilon_{tension}$ is the applied tensile strain (Fig. 5a). The ASR for the scatterer based on through-thickness compression, $\varepsilon_{compression}$, can be expressed as:

$$ASR' = \frac{ab(\varepsilon_{compression}\nu + 1)^2 - ab}{ab} = (\nu\varepsilon_{compression} + 1)^2 - 1 \quad (4)$$

where $\nu$ is the Poisson ratio and $\varepsilon_{compression}$ is the applied compressive strain (Fig. 5b). Eq. 3 and 4 indicate that despite the expansion in the in-plane direction, the through-thickness compression presents a smaller space requirement than the in-plane tension with the same strain magnitude if the Poisson ratio is lower than 0.41 (Fig. 5c). The Poisson ratios can be readily tailored to be close to or even lower than 0, as in auxetic structures by introducing micro/nanoscale pores into the elastomers[64–67]. As a proof-of-concept, a series of models featuring different porosities and pore morphologies were built (Fig. 5d and Supplementary Fig. 17). Specifically, Models #1, #2, and #3 possessed

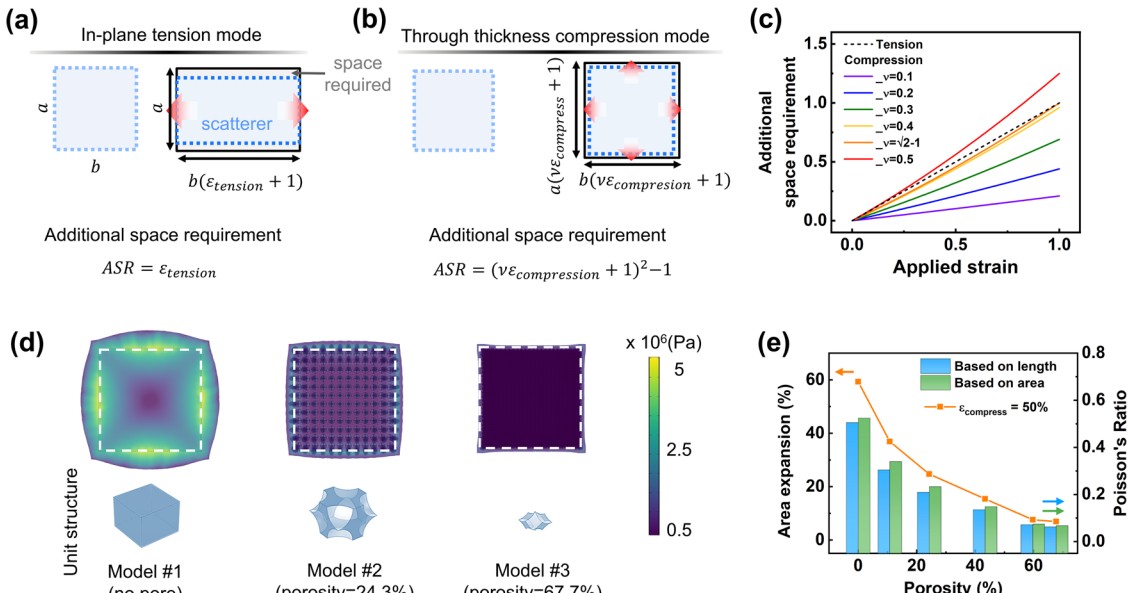

**Fig. 5 | Potentials of the through-thickness compression mode in minimizing the space requirement of mechano-responsive scatterers.** Schematics illustrating the calculation of the additional space requirement in scatterers activated by (**a**) in-plane tension and (**b**) through-thickness compression. *a* and *b* are the initial dimensions of a scatterer, $\varepsilon_{compression}$ and $\varepsilon_{tension}$ refer to the strain of compression and tension. Shaded regions outlined with blue dashed line refer to the shape of scatterer without or under loading, regions outlined with solid black line refer to the required space during the whole operational process. Thin black arrows indicate scales and thick red arrows indicate deformations. **c** Additional space is required due to area expansion of materials with different Poisson ratios upon uniaxial tension/normal compression. **d** Simulated stress distribution of models with varying pore characteristics upon 50% compressive strain. **e** Simulated area expansion and Poisson ratio of the models with different pore characteristics. Poisson ratios were calculated by the relative change of lateral size and area. Source data are provided as a Source Data file.

no pores, sparse closed pores, and dense open pores, respectively. The models present largely disparate deformations upon application of 50% compressive strain (Fig. 5d). Due to the high Poisson ratio of ~0.5 of the modeled elastomeric material like PDMS, the nonporous Model #1 expanded significantly in all in-plane directions, leading to an area expansion of ~60% (Fig. 5e). In contrast, Model #3 exhibited a marginal expansion of less than 10%. Poisson ratios were calculated based on the changes in lateral length and the area expansion (Eq. 4). Regardless of the calculation method, the area expansion consistently decreased with the Poisson ratio, verifying the potential of the proposed through-thickness mode in addressing the long-lasting problem of non-trivial area expansion (Supplementary Movie 3). By optimizing the parameters such as pore geometry, orientation, size, volume fraction, and distribution, the Poisson ratio of porous structures can be effectively reduced, requiring almost negligible extra space for the operation of the scatterers.

## Discussion

This paper introduces a mechano-responsive optical membrane activated by through-thickness compression. We demonstrate that altering the activation mode to through-thickness effectively addresses the fundamental challenges requiring large extra space when conventional in-plane loading mode activates the optical scatterers. This solution is based on successfully decoupling the space requirement from the lateral dimension of the scatterer. Meanwhile, none of the optical performance, simplicity of material composition, or fabrication process of the scatterer is compromised but can be satisfied at the same time.

To fully explore the design principle of the scatterers, a series of 3D nanostructures with inclined pores are fabricated using a 3D patterning technique, each with a single exposure. Both experiments and multiphysics modeling analyses reveal that the pores inclined against the normal direction hold advantages over the vertically aligned pores regarding multiple scattering boundaries in the released state and

faster pore closure under compression. The transmittance of the scatterer with inclined 3D structures can be modulated by a remarkable 89% by through-thickness compression of only 0.05 mm. Layering three ultrathin scatterers yields a high transmittance of 95% and an unprecedented contrast of 94%. Moreover, the through-thickness compression mode offers unique flexibility to control the local transmittance over the entire scatterer selectively, which is unattainable using the conventional in-plane tension mode. Potentially, sensitivities to electric fields, heat, and light can be incorporated into the current system while preserving the current mechano-responses if the corresponding active materials are introduced.

In conclusion, the 3D optical scatterer activated by through-thickness compression and its asymmetric design demonstrated here represent the essence of mechanical metamaterials and may offer a new driving force for the future development of not only smart windows but also miniaturized optic devices, transparent displays, and flexible sensors.

## Methods

### Fabrication of phase masks for patterning with slanted exposure

The fabrication process was based on our previous reports with modifications[68]. Specifically, polyurethane acrylate (PUA, MINS 311rm) was dropped on top of the silicon master and spread out with a polyethylene terephthalate (PET) film. After curing, the PUA/PET was removed from the substrate and used as the template for the second replication. The PUA/PET film was attached to a petri dish, and the prepolymers of hard PDMS (VDT-731 as base and HMS-301 as a cross-linker, Gelest) were spin-coated onto the film at 2000 rpm and cured. Then, soft PDMS (Sylgard 184, Dow) was poured into the petri dish. To make the phase grating slanted, the petri dish was tilted at the required angles that determined the incident angle when curing the PDMS. After curing, the phase mask with the pillar arrays of hard PDMS and a backing layer of soft PDMS was peeled off from the PUA/PET film and cut into the shape of a triangular prism for patterning. The phase-

modulating region of the mask consisted of square-arrayed microholes with a diameter of 480 nm, a relief depth of 400 nm, and a periodicity of 600 nm.

## Fabrication of 3D photoresist templates and 3D scatterers

The procedures to fabricate oblique 3D structures were similar to those used to fabricate normal 3D structures using PnP in terms of exposing, baking, developing, rinsing, and drying[20,54]. It should be noted that the conformal contact between the phase mask and the photoresist (NR5-6000 and NR7-80P, Futurrex) could not only affect the quality of the fabricated structures but also determine whether patterning was possible when the oblique exposure angle was higher than 45° because of the potential total internal reflection. UV light (355 nm) with p-polarization was used for the exposure, during which, a baffle was placed under the phase mask to reduce the haze generated from the mask edges. To prepare the porous PDMS with 3D structures, 0.27 g of PDMS prepolymer (Sylgard 184) was cast onto the template on a 28 mm × 28 mm square-shaped glass substrate. The infiltration of PDMS into the template was conducted with the assistance of a vacuum. When preparing the ultra-thin (46 $\mu$m) but optically dense 3D PDMS, the PDMS prepolymer was spin-coated onto the photoresist template. The spinning speed of 2000 rpm was chosen to guarantee the 3D structure of the photoresist was covered (Supplementary Fig. 12b). After curing the PDMS at 65 °C for 2 h, the template was removed by immersing it in resist remover (RR41, Futurrex) and rinsed in DI water. The as-prepared porous 3D PDMS was used as mechanoresponsive 3D scatterers for light scattering.

## Mechanical simulations

LS-DYNA, a nonlinear finite-element analysis platform with a large library of material models and element formulations, was employed to analyze the strain distributions in the 3D scatterers with varying pore orientations (see the Supporting Information for details). The simulation of areal strains produced by models with different pore characteristics was conducted using COMSOL Multiphysics software. The intrinsic Poisson's ratio of the modeled material was set to be 0.49.

## Optical simulations

Optical simulations for calculating the scattering efficiency and its contribution to the transmittance drop were conducted using commercial FDTD simulation software, Lumerical (see the Supporting Information for details).

## Characterization

Unless otherwise stated, the transmittance in this manuscript refers to normal transmittance which indicates the intensity ratio of the transmitted part of the incident light without being scattered. The transmittance was characterized using an ultraviolet-visible-near infrared spectrophotometer (SolidSpec-3700, Shimadzu) equipped with an integrating sphere, which was used to quantify the optical modulation. Five samples were measured for each case of structural inclination. The refractive index of the photoresist and PDMS was measured using an ellipsometer (ALPHA-SE). An SEM (S-4800, Hitachi) was used to characterize the inclined 3D structures of the photoresist templates and the 3D PDMS. The quasi-static compression tests were performed on a universal tensile machine (ESM303, Mark-10) at a loading rate of 0.5 mm min$^{-1}$.

To measure the normal transmittance under pressure, the 3D scatterer was cut into 10 mm × 10 mm and placed between two glass slides. The glass slides were compressed by a homemade module consisting of transparent acrylic boards driven by precision screws on the edges. The distance between opposite screws was 25 mm. The acrylic boards with a relatively large thickness of 5 mm were used to minimize the bending of the boards during compression. For the normalization of transmittance, a thin layer of PDMS was spin-coated

to cohere the two glass slides so that the light scattering effect caused by the additional boundary between the glasses could be eliminated. To characterize the cyclic performance, two PDMS spacers with dimensions of 1 mm × 10 mm × 0.6 mm were attached to two ends of the scatterer to facilitate the pore recovery when the compression was removed.

## Data availability

The authors declare that all data supporting the findings of this study are available within the article and its Supplementary Information file. Source data are provided with this paper.

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

## Acknowledgements

This work was supported by the Creative Materials Discovery Program (NRF-2020M3D1A1110522, S.J.) and National Research Foundation of Korea (NRF) grants (NRF-2022M3H4A1A04068923, S.J.) funded by the Ministry of Science and ICT of the Korean Government (MSIT), and the Postdoctoral Researcher Support Project funded by Korea University (H.C.). In addition, we thank Prof. X.Lu and Prof. C.Qin for fruitful discussions.

## Author contributions

Conceptualization: H.C., S.N., D.C., and S.J., Methodology: H.C., G.C., D.C., G.B., Y.L., K.K., J.F., and H.Z., Investigation: H.C., T.L., S.M., and S.J., Visualization: H.C., G.C., T.L., S.M., K.K., J.S., and J.W.H., Supervision: J.K.K., J.S., J.W.H., and S.J., Writing—original draft: H.C., Writing—review & editing: H.C., J.K.K., J.W.H., and S.J.

## Competing interests

The authors declare no competing interests.
