## [Peer Review File · Nature Communications]

Compression-Sensitive Smart Windows: Inclined Pores for Dynamic Transparency ChangesREVIEWER COMMENTS

Reviewer #1 (Remarks to the Author):

The manuscript titled "Compression-Sensitive Smart Windows: Inclined Pores for Dynamic Transparency Changes" introduces a novel approach that tackles the significant challenge of the substantial spatial needs associated with mechanochromic smart windows. Out-of-plane compression is proposed to replace the commonly used in-plane tension as the activation mode to minimize the space needed. The discussion on how this new activation mode inherently decouples the spatial requirement, namely the activation deformation and additional area occupied during operation, from the original lateral size of the membrane could be inspiring for the following work in the future.

The as proposed optical membrane is a 3D nanostructured porous PDMS, with the pore orientation regulated by a modified 3D patterning technique controlling the incident light angle to the phase grating. The influence of pore orientation ranging from 0° to 40° in the porous structure on transmittance sensitivity to compression is studied through experiments and well supported by Multiphysics simulations. The results are intriguing, given the high transmittance contrast of 94% from a membrane with a micron-scale thickness, the impressive durability to 100,000 loading/releasing cycles, and the capability to modulate transparency selectively in any area as required. Overall, the work presented in this manuscript is interesting and deserves to be published with some modifications to the issues mentioned below.

1. Although the manuscript focuses on normal transmittance, it is necessary to show the total transmittance and reflectance of the optimized membrane to better evaluate its modulation performance.
2. It is revealed that the sensitivity of the transmittance to compression increases with the rise of the inclined angle of the pores. Can the current 3D patterning technique fabricate pores oriented horizontally? What is the maximum area that can be obtained in a single run using the 3D patterning technique used by the author, and what other issues need to be addressed if large-scale production is desired?
3. It is noted that the model used in the mechanical simulation has a high aspect ratio. Is the model stable under compression? Related descriptions of the modeling conditions should be provided.
4. The choice of materials plays a crucial role in the functionality of the scatterer. What were the key criteria for selecting PDMS? How do the optical and mechanical properties of these materials influence the design and performance of the smart window?
5. Is there potential for incorporating additional functionalities into the smart window, such as UV filtration and sensitivity to other stimuli? How might these be integrated without compromising the window's optical performance?

Reviewer #2 (Remarks to the Author):

In this manuscript, the authors present compression-sensitive smart windows by controlling the closing and opening of pores within the 3D porous structure. This is a very simple and effective strategy. This reviewer suggests a minor revision before it can be accepted for publication.

1. This following reference (Adaptive fluid-infused porous films with tunable transparency and wettability. *Nature Materials*, 2013, 12(6): 529-534) should be cited and the authors should elaborate on the differences of the two similar methods.
2. The influence of factors such as the shape, size, angle, and thickness of the pores should be clearly explained.
3. The compression percentage should be added in Figure 3c and 3j.

REVIEWER COMMENTS

Reviewer #1 (Remarks to the Author):

General Comments: *The manuscript titled "Compression-Sensitive Smart Windows: Inclined Pores for Dynamic Transparency Changes" introduces a novel approach that tackles the significant challenge of the substantial spatial needs associated with mechanochromic smart windows. Out-of-plane compression is proposed to replace the commonly used in-plane tension as the activation mode to minimize the space needed. The discussion on how this new activation mode inherently decouples the spatial requirement, namely the activation deformation and additional area occupied during operation, from the original lateral size of the membrane could be inspiring for the following work in the future.*

The as proposed optical membrane is a 3D nanostructured porous PDMS, with the pore orientation regulated by a modified 3D patterning technique controlling the incident light angle to the phase grating. The influence of pore orientation ranging from 0° to 40° in the porous structure on transmittance sensitivity to compression is studied through experiments and well supported by Multiphysics simulations. The results are intriguing, given the high transmittance contrast of 94% from a membrane with a micron-scale thickness, the impressive durability to 100,000 loading/releasing cycles, and the capability to modulate transparency selectively in any area as required. Overall, the work presented in this manuscript is interesting and deserves to be published with some modifications to the issues mentioned below.

Our response to General Comments: We appreciate the reviewer for these positive comments on our work and the recommendation to publish in Nature Communications.

Comment 1: *Although the manuscript focuses on normal transmittance, it is necessary to show the total transmittance and reflectance of the optimized membrane to better evaluate its modulation performance.*

Our response to comments 1: Thank you for the constructive suggestion.

In addition to the modulation of normal transmittance manifested in the manuscript, we have conducted additional experiments to characterize the total transmittance and total reflectance of the 3D scatterer with optimum structural inclination in the released and compressed state. As shown in Fig. R1, when 10% compressive strain was applied to the scatterer, the total transmittance at the wavelength of 550 nm increased from 77% to 95% while the total reflectance reduced from 23% to 4%. It could be noted that although the normal transmittance was only 6% in the released state (Fig. 3b), the total transmittance remained relatively high which can be rationalized by Mie scattering, granting the scatterer great potential in smart window which diffuses the concentrated sun glare and improves the overall indoor illumination conditions during daytime (*ACS Nano* **2022**, 16, 1, 68–77). Introduction has been modified to highlight the light redistribution of light-scattering membrane.

Fig. R1. Total transmittance and total reflectance of the optimized 3D scatterer before and after the application of 10% total strain.

Modification to the manuscript:

Page 2:

“As an important category in smart windows, light-scattering smart optical membranes can be classified based on the stimuli that activate their optical modulation. For example, mechano-responsive membranes enable tunable transmission and redistribution of transmitted light through

strain-controlled light scattering.”

Page 9:

“In the case of total transmittance and total reflectance, the modulation was approximately 20% but in opposite directions (Fig. S10). The relatively high total transmittance of 77% in the released state can be rationalized by Mie scattering, which typically leads to a high proportion of forward-scattered light.”

Page 12:

“Thanks to the remarkable modulation capability of normal transmittance along with the relatively high total transmittance, the scatterer can also function as diffusers for concentrated sun glare and improve the overall indoor illumination conditions during daytime (20).”

Supplementary Information page 11:

“

Figure S10. (a) Total transmittance and (b) total reflectance of the optimized 3D scatterer before and after the application of 10% total strain.”

Comment 2: It is revealed that the sensitivity of the transmittance to compression increases with the rise of the inclined angle of the pores. (1) Can the current 3D patterning technique fabricate pores oriented horizontally? (2) What is the maximum area that can be obtained in a single run using the 3D patterning technique used by the author, and (3) what other issues need to be addressed

if large-scale production is desired?

Our response to comments 2: Thank you for the thoughtful questions.

(1) 3D structures with horizontal pores cannot be fabricated by simply further increasing the tilting angle. This is because of the exponentially reduced intensity of the transmitted light through the phase mask and the compromised patterning contrast as the incident angle increases (Fig. S4 and S5). The pattern typically displays constructive and destructive interference close to the grating's raised and recess, recurring along the direction of light propagation at a distance defined as Talbot distance Z_T (*Nano Res.* **2021**, 14 (9), 2965-2980.),

$$Z_T \approx \frac{2p^2}{\lambda}$$

where λ and p are incident wavelength, and periodicity of the phase mask. When Z_T is maintained while the radius of the phase modulating raised feature increases, the aspect ratio of the diffraction pattern unit can be easily tailored. This possibility is demonstrated through additional optical simulation to examine the diffraction pattern originated from the gratings with different radii of unit (Fig. R1). The mask design for the 3D structure with horizontally aligned pores can be optimized through inverse calculation (*Sci. Adv.* **2022**, 8, eabm6310). Based on the influence of structural inclination revealed in this work, the development of 3D scatterers featuring even faster pore closure could be a promising future direction.

Radius= 1.7r 1.4r r

Fig. R1. Diffraction pattern originated from phase gratings with different radius of the raised feature.

(2) The large-area scatterer fabricated based on the proximity-field nanopatterning (PnP) technology has been demonstrated in a series of our previous works (*Adv. Sci.* **2020**, 7 (11), 1903708; *ACS Nano* **2022**, 16 (1), 68-77; *ACS Nano* **2020**, 14 (9), 12173-12183). The production area could range from 110 to 300 cm². Uniform optical properties were presented across the samples. The achievable size of the 3D photoresist template has no upper limit by exploiting the sequential multi-exposure technique, and neither does the 3D PDMS (*Nano Res.* **2021**, DOI: 10.1007/s12274-021-3428-6). However, the device size can be strongly restricted by the process required to produce the intermediate layer, which was necessary in the previous studies, as shown in Fig. S6 (*Adv. Sci.* **2020**, 7, 1903708; *ACS Nano* 2020, 14 (9), 12173-12183. *ACS Nano* **2022**, 16, 1, 68–77). Therefore, the fabrication process in this work, which eliminates the need to generate the intermediate layer, shows important progress in developing advanced scatterers that are easy to scale up.

(3) Integrating the PnP technique with a continuous exposure method may realize 3D patterning in a roll-to-roll form and break the current limitation of production dimension and time. By mass production, the 3D patterning would become cost-effective and can be widely exploited in various applications such as optical membranes, sensors, and energy storage devices to improve device performances based on structural effects. This may involve systematic process optimization and re-design of the phase mask geometry. Specific issues include the following:

I. To increase the production area from a single exposure, larger phase masks need to be prepared, and the light beam for exposure should be expanded while ensuring its uniformity and coherence.

II. Alignment between the phase mask and photoresists is critical to obtaining uniform patterns. Consistent physical conformal contact should be guaranteed during the adhesion/decohesion process throughout continuous fabrication.

III. Uniformity of the exposure dose should be taken into consideration when the production is conducted in either a continuous or a sequential manner.

Related discussion has been incorporated into the manuscript:

Page 15:

“Future directions could include designing optimized geometries, such as auxetic structures, for efficient pore closure and extending 3D patterning techniques to realize these structures through inverse calculations.”

Supplementary Information page 7:

“The device size can be strongly restricted by the process required to produce the intermediate layer, which was necessary in the previous studies, as shown in Fig. S6 (20, 27). Therefore, the fabrication process in this work, which eliminates the need to generate the intermediate layer, shows important progress in developing advanced scatterers that are easy to scale up.

To further scale up the fabrication, integrating the PnP technique with a continuous exposure method may realize 3D patterning in a roll-to-roll form and break the current limitation of production dimension and time. By mass production, the 3D patterning would become cost-effective and can be widely exploited in various applications. This may involve systematic process optimization and re-design of the phase mask geometry, where issues include enlarging the phase mask and light beam, guaranteeing the consistent alignment between the mask and photoresist, and ensuring the exposure dose across the large-area sample.”

Comment 3: *It is noted that the model used in the mechanical simulation has a high aspect ratio. Is the model stable under compression? Related descriptions of the modeling conditions should be provided.*

Our response to comments 3: Thank you for your thoughtful comment. In our finite element numerical simulation, we applied periodic boundary conditions to the unit cell, as illustrated in Fig. R2(a), to eliminate edge effects (Fig. R2b). Consequently, the boundaries of the unit cell are

constrained such that the boundary nodes on each face exhibit the same displacements. This setup effectively simulates the behavior of the developed material under a compression load applied in the -z direction, ensuring stable responses.

Figure R2. Boundary condition setting of the mechanical simulation. (a) Periodic boundary conditions applied to a unit cell selected from the interior of the material. (b) Deformations of the porous component when no periodic boundary is applied.

Modification to the manuscript:

Supplementary Information page 14:

Fig. S13.

(a) Description of the FEA model. (b) Periodic boundary conditions applied to a unit cell selected from the interior of the material. (c) Buckling behavior of the porous component when no periodic boundary is applied.

In the finite element numerical simulation, a periodic boundary condition was applied to the unit cell, as illustrated in Fig. S13b, to eliminate edge effects such as buckling (Fig. S13c). This condition constrains the boundary nodes on each face to exhibit the same displacements, effectively simulating the behavior of the developed material with a periodic porous structure under a compression load applied in the -z direction, ensuring stable responses of the simulation.”

Comment 4: *The choice of materials plays a crucial role in the functionality of the scatterer. What were the key criteria for selecting PDMS? How do the optical and mechanical properties of these materials influence the design and performance of the smart window?*

Our response to comments 4: Thank you for your thoughtful question. The opaqueness of the scatterer in the released state originates from the contrasting refractive indices between the pore and the matrix material, while the transparency in the compressed state is determined by the intrinsic transmittance of the matrix. Therefore, for a high-performance scatterer that can repeatedly respond to mechanical stimuli: mechanically, the matrix material should possess high elasticity and low adhesiveness for recoverability, robustness for durability, and low hysteresis for fast response; optically, the material should be of high transparency to visible light for the high transmittance achievable as well as the modulation range. PDMS is known to be highly elastic, non-adhesive, mechanically robust, and highly transparent to visible light and thereby selected as the matrix material in this work.

Modification to the manuscript:

Page 8:

“PDMS was selected as the matrix material considering its elasticity, non-adhesiveness, mechanical robustness, and high transparency to visible light.”

Comment 5: *Is there potential for incorporating additional functionalities into the smart window, such as UV filtration and sensitivity to other stimuli? How might these be integrated without compromising the window's optical performance?*

Our response to comments 5: Thank you for considering the prospect of this work.

(1) The functionality of UV filtration could be introduced by incorporating nanoparticles (NPs) that can absorb or scatter UV light. NPs with a wide bandgap such as ZnO, TiO₂, and carbon-based nanodots can effectively absorb and scatter UV light when blended into PDMS (*J. Phys. Chem. C* **2018**, 122, 22, 12114–12121). Particles smaller than tens of nanometers can scatter the UV while

avoiding unwanted disturbance of visible transmittance. Organic absorbers, including benzotriazole, could also be considered (*Polymer* **1985**, 26 (9), 1288-1296.), but they may be less optimal due to their poor stability under UV. Another approach is to use photoluminescent materials that can convert the incident UV to light at other wavelengths, which could not only filter UV (*Sci. Adv.* **2023**, 9 (7), eade2585), but also increase the intensity of the transmitted visible light.

(2) Replacing the PDMS with other functional elastomers such as polyvinylidene fluoride and polyurethane with high dielectric constant could incorporate sensitivity to an electric field. When transparent electrodes are coated on the top and bottom surface, the dielectric polymers could be compressed upon an electric field and close the pores, turning the smart membrane from opaque to transparent. In addition to electrochromic properties, thermochromic and photochromic properties can be introduced if corresponding active materials including V₂O₅, spiropyran, and Poly(N-isopropylacrylamide) are exploited. However, it should be noted that as long as the material system fulfills the mechanical and optical requirements mentioned in our responses to the last question, the existing mechano-responses shall be maintained.

Modification to the manuscript:

Page 17:

“Potentially, sensitivities to electric fields, heat, and light can be incorporated into the current system while preserving the current mechano-responses if the corresponding active materials are introduced.”

Reviewer #2 (Remarks to the Author):

General Comments: *In this manuscript, the authors present compression-sensitive smart windows by controlling the closing and opening of pores within the 3D porous structure. This is a very simple and effective strategy. This reviewer suggests a minor revision before it can be accepted for publication.*

Our response to General Comments: We appreciate the reviewer for these positive comments on our work and the recommendation to publish in *Nature Communications*.

Comment 1: *This following reference (Adaptive fluid-infused porous films with tunable transparency and wettability. Nature Materials, 2013, 12(6): 529-534) should be cited and the authors should elaborate on the differences of the two similar methods.*

Our response to comments 1: Thank you for the suggestion. Our work proposes a porous elastomer membrane that transitions from initially opaque to transparent upon through-thickness compression, due to the closure of pores that act as scattering sites. In contrast, the referred work utilized a liquid-infused porous matrix on an elastic substrate, where the liquid overweted the porous matrix forming a smooth surface coating in the relaxed state and stretching caused the liquid to redistribute and expose air pockets, transitioning from initially transparent to opaque.

Our novel activation mode—through-thickness compression instead of in-plane tension—along with optimized pore geometry and prudent design of the material system regarding refractive index, allows us to address the fundamental challenge of high spatial demand during operation and achieve the best optical modulating performance to date. By comparison, as a pioneering work from ten years ago, the referenced study was novel for demonstrating active control of not only transmittance but also wettability by strain and temperature.

The recommended reference was cited as reference 26. However, since this work was a solid step of progress in the field of mechano-responsive smart membranes, additional discussion is now incorporated in the Introduction.

Modification to the manuscript:

Page 3:

“Two-dimensional (2D) scatterers achieve adaptive light scattering based on reversible flattening of surface wrinkles and cracks or infusion of surface roughness by liquid (26, 30-45). Although 2D

scatterers can be easily fabricated, their optical contrast is lower than 60%, and the strain required for operation is usually higher than 30%, resulting in limited applicability (30-45).”

Page 24:

“26. Yao, X. Adaptive fluid-infused porous films with tunable transparency and wettability. *Nature Materials* 12, 529–534 (2013).”

Comment 2: *The influence of factors such as the shape, size, angle, and thickness of the pores should be clearly explained.*

Our response to comments 2: Thank you for your thoughtful question.

Shape and angle: The influence of pore shape is paramount. As discussed in this manuscript, the angle, namely the orientation of pores, is important for elliptical pores, making the pore closure facilitated as the angle to the normal direction increases. This argument has been corroborated by multiphysics simulation (Fig. 4b-4e, Fig. S15). When not limited to ellipsoids, the shape of pores can significantly affect pore closure behavior under compression. In particular, the pores in auxetic structures, which have a negative Poisson ratio, are easy to close thanks to the well-designed rotation at the structure hinges. Designing optimized geometry for efficient pore closure and extending the 3D patterning technique to realize these structures by inverse calculations could be our future direction.

Size: Although the influence of the initial pore size is beyond the scope of this work, it could be an important parameter to study in the following works. Based on existing studies, it can be noted that pore size is one of the dominant factors determining the scattering performance (*Applied Optics* **1980**, 19, 1505). A particle’s scattering area increases with its increasing physical area (*Application of Light Scattering to Coatings: A User’s Guide, Springer, 2014*). The scattering efficiency of a particle, defined as the scattering area divided by the physical cross-sectional area of the particle, increases drastically with the growth of particle diameter, and fluctuates when the diameter becomes the same order as the incident wavelength (*Application of Light Scattering to Coatings: A User’s*

Guide, Springer, 2014). Therefore, there lies an optimum size for pores to maximize the scattering efficiency and thus opaqueness. The transmittance contrast of the scatterer can be optimized. Moreover, it was also indicated that when the scattering efficiency is at its peak, the decrease in the scattering site's diameter brings about a rapid drop in the efficiency, as the “fluctuation region” is skipped. This means that the sensitivity and contrast of a pressure-induced scatterer can be greatly improved simultaneously.

Thickness: As indicated from Fig. 3g, increased thickness of the porous layer renders stronger modulation of transmittance, with the transmittance contrast improved from 89% to 94%. However, as the transmittance in the released state decreases exponentially, the modulation capability saturates as the thickness of the porous layer rises beyond 36 μm . Excessive thickness could lead to additional light absorption which compromises the maximum transmittance and thus the achievable contrast.

Related discussions are incorporated in the manuscript:

Page 11:

“Interestingly, however, the compression-mode 3D scatterer developed here presented almost identical maximum transmittance in the compressed state among all samples with different numbers of layers ranging from 1 to 3 (Figs. 3g and S12c). The negligible additional absorption can explain this observation due to the ultrasmall overall thickness and the elimination of interfaces through thickness compression.”

“Therefore, increasing the thickness of the patternable submicron 3D structure would be a promising future direction to achieve improved scattering performance using one single layer of scatterer (50).”

Page 15:

“Future directions could include designing optimized geometries, such as auxetic structures, for efficient pore closure and extending 3D patterning techniques to realize these structures through inverse calculations.”

Page 16:

“By optimizing the parameters such as pore geometry, orientation, size, volume fraction, and distribution, the Poisson ratio of porous structures can be effectively reduced, requiring almost negligible extra space for the operation of the scatterers.”

Comment 3: *The compression percentage should be added in Figure 3c and 3j.*

Our response to comments 3: Thank you for the constructive suggestion. Figure 3c and 3j have been modified to indicate the compressive strain applied.

Modification to the manuscript:

Page 33:

“

”

REVIEWERS' COMMENTS

Reviewer #1 (Remarks to the Author):

The authors have made revisions based on the reviewer's comments, and I have no further comments on the revised manuscript. I believe that the revised version can be accepted for publication.

Reviewer #2 (Remarks to the Author):

The authors have addressed my questions and comments.

REVIEWER COMMENTS

Reviewer #1 (Remarks to the Author):

The authors have made revisions based on the reviewer's comments, and I have no further comments on the revised manuscript. I believe that the revised version can be accepted for publication.

Our response to General Comments: We appreciate the reviewer for the positive comments on our work and the recommendation to publish in *Nature Communications*. We believe the manuscript has been improved thanks to the constructive suggestions from the reviewer.

Reviewer #2 (Remarks to the Author):

General Comments: *The authors have addressed my questions and comments.*

Our response to General Comments: We appreciate the reviewer for the time and patience to review our work. We believe the manuscript quality has been improved thanks to the constructive suggestions from the reviewer.